# Building the Foundation of Aquatic Literacy in 4–6 Years-Old Children: A Systematic Review of Good Pedagogical Practices for Children and Parents

**DOI:** 10.3390/ijerph19106180

**Published:** 2022-05-19

**Authors:** Léa Mekkaoui, Christophe Schnitzler, Michel Sidney, Joseph Gandrieau, Fabien Camporelli, François Potdevin

**Affiliations:** 1Univ. Lille, Univ. Artois, Univ. Littoral Côte d’Opale, ULR 7369—URePSSS—Unité de Recherche Pluridisciplinaire Sport Santé Société, 59000 Lille, France; michel.sidney@univ-lille.fr (M.S.); joseph.gandrieau@univ-lille.fr (J.G.); francois.potdevin@univ-lille.fr (F.P.); 2Unité de Recherche 1342, Faculté des Sciences du Sport de Strasbourg, Université de Strasbourg, 67081 Strasbourg, France; cschnitzler@unistra.fr; 3CLERSE, UMR 8019, Centre National de la Recherche Scientifique (CNRS), 59800 Lille, France; fabien.camporelli@univ-lille.fr

**Keywords:** swimming intervention, can swim, parental supervision, preschoolers

## Abstract

Children between the ages of 4–6 years represent the population most affected by drowning accidents, while their early involvement in physical activity, and more specifically in aquatic activities is a key factor in their future physical life journey. The systematic review’s purpose was to identify aspects in the intervention’s studies with children and/or their parents that had a significant impact on the Aquatic Literacy (AL) dimensions mentioned as motor, psychological, affective, and cognitive. The PICO method was used to define the research question and PRISMA checklist searched for articles in nine databases: Cochrane, Embase, ERIC, ProQuest, PsychInfo, PubMed, Scopus, SportDiscus, and Web of Science. Eligibility criteria were: (1) English language, (2) primary research, (3) population of 4–6 year old children or their parents, (4) intervention study design, and (5) results related to at least one of the AL domains. The strength of evidence and the risk of bias were assessed. Results showed relatively poor number of studies for such a vulnerable population regarding the drowning risk (*n* = 8 for parents and *n* = 14 for children intervention). Studies did not show a consensus on which educational approach was more beneficial than others. Concerning parental education, results were rather homogeneous, especially concerning the theoretical frameworks employed and the relevancy to include parents in swimming programs. The development of pedagogical tools for promotion and evaluation, based on the AL theoretical framework, could help to clarify the question of “how to teach” children to prevent drowning and engage young children in long-term physical activities.

## 1. Introduction

Words such as “diving”, “swimming”, “floating”, and “gliding” are terms evoking the pleasure humans can experience from aquatic pursuits. Indeed, it comes as no surprise that different forms of aquatic recreation figure prominently amongst our favorite pastimes [1,2]. In view of the sedentary lifestyle and the lack of physical activity (PA) observed for many years among young people [3], water can be a rich and abundant medium for PA, allowing a sustainable commitment to an active and healthy lifestyle [4]. Aquatic activities are often promoted for their health benefits [5,6,7,8], and positive childhood experiences can have a significant impact on lifelong engagement with them [9], encouraging educators to support maximum enjoyment and confidence in children in these environments. However, these benefits can be tragically overshadowed if children are not empowered against the possibility of drowning. This “dark side” of water recreation is partly responsible for the deaths of more than 365,000 people worldwide every year, half of whom are children and young adolescents [10]. Among the victims, the 4–6-year-old age group is over-represented, making them more vulnerable in, on, and around the water [11]. According to the World Health Organization (WHO), tragedy is not inevitable in aquatic activities and could be avoided by a series of political, structural, and pedagogical measures. Amongst these, high-quality aquatic education, from an early age onwards, appears to be key for empowering participants to be protected from drowning [12].

The questions of protective motor skills against drowning accidents in young children, and that of the optimal age to start teaching children to swim, have been widely explored in the literature. The question of an optimum age to start learning to swim has been framed by different theoretical frameworks [13].

The first studies which focussed on an optimal age for learning to swim were first framed within maturation theories [14] of motor development. These studies investigated which aquatic skills could evolve through chronological time inherent to maturation of the central nervous system. Mead [15] and McGraw [16,17] were the pioneer researchers in this field, identifying the ages at which children first acquired aquatic skills. Empirical theories were based on an opposing assumption: considering that the acquisition of aquatic skills was prominently dependent on previous water experiences. This view, strengthened by an environmentally based constructivism model, is most prevalent in training curricula, regardless of the institution (school, Red Cross, YMCA, sports federations, etc.). They advocated that new motor skills could be learned by continuous interactions with aquatic environments, according to specific identified stages (for example, small ‘dog paddling’ movements precede longer backward trajectory actions, or ‘leg pedaling’ movements precede more sophisticated kicking actions). The pedagogical challenge in that approach was not to determine an optimal age for “being able to learn to swim”, but rather consisted of situating each learner at their appropriate level, in order to offer them the most relevant experiences to progress to a higher stage. Finally, the most recent model structuring the vision of aquatic skills acquisition is framed by an Ecological Dynamics model, and specifically by the Constraints-Led approach (CLA) [18,19]. Learning to swim is considered as potentially transient behaviors that emerge from interactions between three dimensions: individual characteristics (e.g., age, experience level or body composition), the perceived aquatic task goal(s) (e.g., treading water, economic forward motion, immersion, etc.), and the physical or social environment (negotiating a pond, river, or rough sea conditions, in warm or cold weather). From this perspective, use of subjective risk scenarios, in which learners must adapt spontaneously (e.g., falling backwards, unstable standing conditions, encountering unexpected objects, needing to find an exit from a body of water) develops adaptive skills and facilitates mobilising learners’ motor, cognitive, and affective resources to consolidate aquatic skills [20].

The question of optimal age also raises an issue of educational responsibility [21]. Supporting children to become more confident as early as possible around water can lead to a greater “attraction” towards aquatic environments, facilitating PA, but increasing exposure to risks of accidents, loss of confidence, and drowning. Moreover, Morrongiello et al. [22] showed that, as children aged 2–5 years progressed through lessons, the perception of danger and parental supervision tended to decrease, making it essential to include parents in the process of children’s aquatic education. Many institutions, such as the American Academy of Pediatrics [23], have not advocated early training, because of a belief that children do not display attitudes and knowledge to behave responsibly around water. Various case studies [24,25,26] have suggested that learning before the age of 3 years, may enhance future learning. On the contrary, Parker and Blanksby [27] and Anderson and Rodriguez [21] have suggested that progress may occur faster between the ages of 4 and 6 years, without previous readiness. These findings have led the American Red Cross Scientific Advisory Council in 2019 to counsel against the standardisation of swimming lessons before age 4 years.

This intertwining of theoretical frameworks is also apparent in didactical research focusing on “what to teach” to prevent young children from drowning. This lack of clarity in theoretical framing was highlighted by Stallman [28] who proposed the concept of “Water Competence” to bypass the epistemological debate about the definition of “can swim” and to focus on “what needs to be taught” during swimming lessons. This heterogeneity of theories underlying existing research is reflected in the multiplicity of concepts used. A first level of conceptualisation is anchored exclusively in the dimension of motor skills. There is a noticeable frequency of typical notions in the scientific and pedagogical literature such as “swimming ability”, “swimming skills”, “aquatic skills”, “survival skills”, “rudimentary skills”, “water readiness”, “pre-requisite skill level”, “swimming proficiency” [29]. This terminology is focused on behavioural indicators regarding what needs to be learned in order to move safely in water. A second level of concepts is used, such as “drownproof”, “drowning prevention” and especially the concept of “Water Competence” initiated by Langendorfer and Bruya [26], and, more recently, by Stallman et al. [28]. At this more holistic level of conceptualisation, “Water Competence” highlights an ability to cope with a potential risk of drowning by mobilising motor capacities, but also cognitive (making decisions, knowledge about the aquatic environment and humans around) and affective resources (managing emotions). Stallman et al. [28] identified 15 water competences that could significantly reduce the risk of drowning. The importance of each of these competences was supported by research findings, addressing the question: “what needs to be taught to protect children from drowning”. This proposal is now widely accepted and has inspired many training programs around the world [30,31,32]. A third level of conceptualisation is inspired by the ‘physical literacy’ framework [33] which is defined by “the motivation, confidence, physical competence, knowledge and understanding that an individual possesses and enables him to value and take ownership of his commitment to physical activity throughout their lives” [34]. Physical literacy for aquatic environments or ‘Aquatic Literacy’ (AL) appears very close to the concept of water competence, including positive motivations to engage in water activities [35].

According to Denehy et al. [36], the lack of theoretical framing in aquatic competence research has inhibited the development of a clear understanding of good practice and its underpinnings in physical education. In a meta-analysis, Leavy et al. [37] confirmed this idea by showing that the majority of prevention programs and their assessment have been implemented without a theoretical framework to frame and understand the behaviours of young children or their parents. This deficit makes the overall message difficult to identify. Moreover, if the questions of “optimal age” and “what to teach” have been widely studied by the researchers, the “how to teach” question tends to be based on pedagogical beliefs, rather than evidence-based interventions [38]. Therefore, it is essential to identify the most effective intervention modalities for the most vulnerable population (young children and their parents). This focus will not only help to guide early “how to swim” teaching strategies to develop participant safety, but also would ensure engagement in future aquatic activities throughout the lifespan, forming the foundation for their aquatic literacy.

In this paper, we present an up-to-date systematic analysis of the effectiveness of teaching strategies used for young children (4–6 years old), and their parents, based on empirical data. The aim, designs and methodological quality of the intervention studies are presented and discussed critically. The effects of the interventions are examined on different dimensions (motor, psychological, social, and cognitive). With the information given in this systematic review, we aim to inform and support stakeholders in the field of aquatic education, from curriculum makers to pedagogues, in order to contribute to the overall quality of aquatic education.

## 2. Materials and Methods

This systematic review was carried out by using the protocol developed by Cochrane Institute [39] and following the checklist of the Preferred Reporting Items for Systematic Reviews and Meta-analysis (PRISMA) guidelines [40]. The aim of this systematic review was to identify whether there are evidence-based methods to optimise the development of aquatic literacy in young children.

### 2.1. Literature Identification

Based on the PICO protocol (Participants—Intervention—Comparison—Outcomes), the research question has been formulated (i.e., For pre-school children aged 4–6 years with no specific health conditions and their parents, what characteristics of educational interventions are effective in improving safety and/or engagement in, on and around the water as compared to children from the same age range who did not benefit from those interventions?) and relevant keywords were selected and are presented in Appendix A.

The keywords related to each of the following categories have been added using the boolean operator “OR” and search techniques, such as truncation and/or phrase marks, adapting them to each database: population (e.g., preschool * OR parent *), intervention (e.g., “swim * training” OR “aquatic lesson *”), outcomes (e.g., “tread * water” OR pleasure), generic terms (e.g., drown * OR “aquatic literacy”). These categories were then combined using the boolean operator “AND” and some terms related to certain pathologies were excluded using the “NOT” operator (e.g., depression OR asthma) to obtain the search algorithm. This algorithm was run during between 28 October and 15 November, 2021, on nine online databases: Cochrane, Embase, ERIC, ProQuest, PsychInfo, PubMed, Scopus, SportDiscus, and Web of Science. The specific search strategy for each database explored are presented in Appendix A.

### 2.2. Selection Process

After collecting the results from databases, duplicates were removed. Then, titles and abstracts of each article were screened independently by two researchers (LM and CS). Disagreements on study eligibility were resolved by an external reviewer (FP). Reference lists of studies were also screened for potentially eligible records.

To be included in the full-text analysis, studies were required to satisfy the following criteria: (1) English language, (2) primary research, (3) part of the population between 4- and 6-year old children or their parents, (4) intervention study design, and (5) results related to at least one of motor, psychological, social or cognitive aspects.

The exclusion criteria of studies were: (1) children with disabilities or pathologies, (2) articles which the 4–6-year-old population has not been subdivided, and (3) articles without specifying the age of the children. 

### 2.3. Data Extraction

Relevant information has been extracted from the full-text analysis and then moved to an Excel spreadsheet. The following information were: (1) bibliographic information (title, authors, year of publication), (2) strength of evidence [41], (3) objective, (4) theoretical framework(s), (5) intervention characteristics, (6) sample size and characteristics, (7) measurement tool(s), (8) outcomes (referring to the four physical literacy domains of Keegan et al. [42]), (9) quantitative and qualitative summary according to group or subgroup when available, and (10) risk of bias.

The data were extracted independently by two researchers and an external opinion was solicited from a third researcher (FP) in case of disagreement. 

Studies were categorised according to their measurements towards children exclusively, children and their parents, and parents exclusively.

### 2.4. Level of Confidence and Risk of Bias Assessment

Two tools were used to assess the studies quality: (1) the strength of the evidence grid of Ackley et al. [41] and (2) risk of bias grid adapted from the Cochrane checklist [39].

To assess selection bias, we examined whether the populations were randomised and the percentage of dropouts (<20%). For information bias, we assessed whether the group was comparable at baseline characteristics, if the baseline values were accounted for, and if the intervention were blinded for population and examiners. Finally, for the bias analysis, we assessed whether the timing of measurement was comparable between intervention and control groups, adequate statistical procedure and presence of *p*-value, effect size and confidence interval. Two researchers (LM and CS) independently evaluated the study quality according to these two tools. Disagreements on study quality were resolved by an external reviewer (FP).

## 3. Results

### 3.1. Overview

The PRISMA flowchart in Figure 1 shows the process of identification, screening, and inclusion of studies. The search yielded 976 results. After removing duplicates, a total of 838 titles were screened. Of these, 725 were excluded during the title screening, and 54 during the abstract screening phase, leaving 50 studies for full-text assessment. During this stage, 12 more articles were included in the full-text analysis by a snowballing process and screening the bibliography and references from the relevant articles.

Twenty-one studies were selected for this review. Table 1 (1 study [43]), Table 2 (13 studies [44,45,46,47,48,49,50,51,52,53,54,55,56]), and Table 3 (7 studies [22,57,58,59,60,61,62]) show the main characteristics of the interventions in terms of objective, level of confidence, theoretical framework used, population, intervention description, measurement tools, and outcomes domains for studies including children and their parents, only children and only parents, respectively. Figure 2 shows a summary of the AL domains concerned in the outcomes of the studies, and Figure 3 shows the risk of bias analysis.

### 3.2. Main Characteristics of Interventions towards Children Aged 4–6 Years

Among the 21 studies reviewed, there was one instance of a concerned child associated with their parents (Table 1) and 13 instances of concerned children exclusively (Table 2). The samples built for the studies ranged between 12 [56] and 5129 [46] children; the duration of the interventions ranged from 90 min [46] to 2-year weekly lessons [55]; and nine studies were explicitly anchored in a theoretical framework [43,44,45,47,49,50,52,54,55]. Intervention methods were assessed by measuring the effects of different pedagogical and environmental teaching variables: the types of pedagogy (linear pedagogy vs. non-linear pedagogy [43], playful pedagogy [44]; the depth of the swimming pool [45,52], the frequency [49] and amount [47,50,55] of swimming sessions, the use of in-school intervention [46,48,51,53,56] and the use of information technology [54]. Over the 21 studies explored, only 5 used a control group to strengthen the evidence analysis [47,48,50,53,55]. The methodological strategies to conduct the designs included six studies using validated motor tests [43,44,45,47,49,52], two studies using unvalidated motor tests [50,55], one study using validated questionnaires [43], six studies using questionnaires created specifically for the study [46,48,51,53,55,56], and one study using video analysis [54].

### 3.3. Main Characteristics of Interventions towards Parents

Seven studies concerned parents exclusively (Table 3), and one study concerned parents associated with children (Table 1). The samples built for the design studies ranged between 15 [59] and 4010 [60] parents. The duration of the programs is variable: single campaigns [58] up to a 6-week campaign [60], instructions or feedback about children during their swimming teaching ranging from 10 weeks of teaching [62] to 8 months of teaching [22], and training ranging from 20 min [59] to 1 h [61]. Five studies were explicitly anchored in a theoretical framework [43,58,59,60,61]. Intervention methods were assessed by measuring effects of different communicational and pedagogical variables: drowning prevention campaign [58], presence during the children swimming lessons [22,43], effect of close-call drowning [57], receiving feedback about children improvement [57], parental self-instructional programs [59], and parental water safety programs [60,61,62].

Over the eight studies explored, only three used a control group [57,60,61] to strengthen the evidence analysis. Only two studies used validated questionnaires in their experimental design [59,61], four studies used questionnaires created specifically for the study [22,43,57,58,62], and one study used video analysis [60].

### 3.4. Outcomes Explored through the Aquatic Literacy Concept

The outcomes measured in children analysis concerned the motor (*n =* 9), psychological (*n* = 5), cognitive (*n =* 5), and social (*n =* 2) dimensions. Only two studies covered three dimensions: motor, psychological, and social [43,55]. The types of effects measured in the parent studies were cognitive (*n =* 7), psychological (*n =* 3), and social (*n =* 1). Figure 2 synthesizes the domains explored by each studies reviewed.

### 3.5. Strength of Evidence and Risk of Bias Analysis

Regarding the strength of evidence, two articles had evidence from qualitative studies [44,58], nine from case-control or cohort studies [22,46,50,51,53,55,56,59,62], nine from controlled trials without randomisation [43,45,47,48,49,52,54,57,61], and one from a randomised and controlled study [60]. 

Figure 3 shows the adapted risk of bias grid of Cochrane [39], of the 21 articles selected. None had a low risk of bias for both selection, information, and analysis. None presented a low selection risk. Only one article performed a selection by randomisation [60], and six had a dropout lower than 20 percent [43,44,46,49,55,62]. Seven had a lower information bias. Three did not consider the baseline values [44,47,60] and half performed a blinded intervention [45,47,48,49,50,52,53,54,58,59,60]. Regarding the statistical bias, two had a medium risk of analysis [51,62] and four had a high risk [44,55,56,59]. Finally, six articles had low risk of bias for information and analysis but high to medium risk of bias for population selection [45,48,50,52,53,54].

## 4. Discussion

The aim of this systematic review was to identify pedagogical evidence to: (1) improve pedagogical strategies for building the foundations of early childhood aquatic education, and (2), identify intervention strategies for parents to raise awareness and improve their attitudes towards the risk of drowning. The main results showed that: (1) although the literature displays the existence of different pedagogies, capable of improving different dimensions of aquatic literacy, deciding how to teach aquatic skills depends on the teacher’s pedagogical objectives; and (2), parents should be included in the educational programs to maximize drowning prevention.

### 4.1. A Relatively Poor Number of Studies for Such a Vulnerable Population Regarding the Risk of Drowning

Our systematic review of nine international databases showed a limited number of studies that addressed the question of how to teach children to swim in the age range 4 to 6 years. Many epidemiological studies throughout the world have highlighted the vulnerability of this population to drowning [10], providing a legitimate educational question from a scientific perspective. The plethora of learning programs aimed at this age group in different institutional settings confirms Stallman’s point [38] that the pedagogies proposed for use are based more on beliefs than on scientific evidence. While the questions of “when and what” to teach have been deeply explored in the scientific literature, the question of pedagogical methods (how to teach) needs further exploration. This weakness in the applied scientific literature can, in part, be explained by the methodological and epistemological issues involved in research in the contexts of real-life teaching and learning. Carreiro Da Costa [86] highlighted an evolution in the theoretical frameworks used in research on effects of pedagogies used, initially anchored in a positivist process-product paradigm (measuring the effects of one or more variables, while neutralising the effects of others). The impossibility of identifying obvious regularities in samples of teachers and students within educational settings has led researchers in the field of educational interventions to adopt mixed methods approaches. These include seeking evidence of the teacher’s thinking and actions and observing and recording multiple interactions with, and between, students. This complexity of measuring the teaching-learning process in an aquatic environment for children may have limited research projects in this area. Across the 21 studies included in our review, the lack of an explicit theoretical framework (*n* = 8) or the variety of frameworks employed relating to communicating science [58], public health [43,50,60,61], motor learning [43,47,49], psychological development [44,52,55] and water skills acquisition [45,52], corroborate the conclusions of Denehy et al. [36] and Leavy et al. [37]. For them, the heterogeneity, or even the absence of theoretical frameworks, make it difficult to summarise the main findings in order to develop clear and consensual recommendations.

### 4.2. What Are the Criteria for Selecting One Pedagogical Approach, Rather Than Another?

The challenge addressed by our PICO methodology was to identify the pedagogical characteristics that significantly impact the progress of students and their parents, in limiting risks of drowning, while encouraging children to engage in aquatic physical activity throughout life to develop aquatic literacy. Analysis of all the studies focusing on effects of pedagogical characteristics of aquatic programs revealed a high level of heterogeneity in theoretical frameworks implemented measurement tools and data used to assess children’s learning outcomes. Most of the intervention studies we found emphasised a skill- or competence-based approach and helped to question the effectiveness of different pedagogy types on the physical dimension of physical literacy [42]. The results highlighted three main findings: 

In terms of the temporal and quantitative characteristics of the programs, Erbaugh [47] compared the evolution of children’s swimming skills exposed to programs of different duration. They reported that the length of the exposure to interventions was the main driver of swimming skill acquisition. Bradley et al. [49] confirmed this trend by showing similar progress in comparing a massed vs. distributed swimming teaching programme.

Concerning skill acquisition, the use of environmental constraints (shallow vs. deep water learning, use of video feedback) showed a significant effect on skill acquisition and/or improvements in confidence [45,52]. 

The type of pedagogy adopted also provides interesting insights. To exemplify, although most studies proposed the use of competence-based and top-down structured interventions [50], some pedagogies, based on less structured activities and play also improved water-related physical competences [44].

In that regard, the pedagogy selected for use seems to rely on the priorities of the teacher and their intentions to improve the one or more dimensions of aquatic literacy. Invernizzi et al. [43] analysed the effect of two types of pedagogy: one more structured (linear pedagogy, or LP) and another which emphasised less structured learning (nonlinear pedagogy, or NLP). Their results showed that LP improved the reproduction of more specific aquatic skills compared to NLP, but that the latter simultaneously increased the confidence and enjoyment of learners more, stimulating further learning, than LP. Other interventional studies evidenced a combined effect of environmental constraints on combined dimensions of aquatic literacy. Costa et al. [45] showed that shallow water teaching led to the learning of more advanced motor skills, since children felt more confident, which encouraged them to explore new motor possibilities. They highlighted methodological biases by considering that, in deep water, the program could not be delivered in the same way for safety reasons (using more directed pedagogy, and occasional use of flotation devices). Rocha et al. [52] reported similar findings using a similar methodology, with improvements emerging in a broader spectrum of water skills when teaching in shallow water. However, the results showed that these between-group differences in skill acquisition disappeared after 18 months of instruction. These results, which favour teaching in shallow water, are the only ones permitting comparisons of similar outcomes. However, since those effects vanished over time, and the potential biases due to the use of flotation equipment, shows that both approaches (shallow vs. deep water teaching) could be equally relevant according to the learning context (short or long exposure). Arhesa et al. [44], without using a control group, showed that a ‘play’ based pedagogy improved water skills and confidence in young children. Bunker et al. [54] assessed the effects of video feedback on front crawl learning and showed significant effects in flutter skills kick performance compared to the control group.

We also found evidence that the dimensions of physical literacy could be improved, whether related to the cognitive [46,48,51,53,56], psychological [53], or social dimensions [43,55]. Overall, these studies showed significant improvements in all domains investigated and allowed the authors to validate the putative benefits of their respective programs. Taken together, those results showed that many dimensions of physical literacy might be subject to improvement, so it is up to the teacher to define and prioritise their objectives and select a method accordingly. 

However, the overall quality of the study design (poor to high risk of biases), the lack of control-randomised studies and the multiplicity of pedagogical characteristics, and studies with heterogeneous methodologies call for caution in interpreting and applying the results in practice. The data highlight that more research on the topic of aquatic literacy development is needed. 

### 4.3. What Kind of Interventions Addressed to Parents Help Improving Safety and Protection for Their Child?

Our selection of articles highlighted the importance of implementing specific programs to address parental needs, as children’s participation in learning to swim programs might paradoxically expose them to additional risks of drowning.

Several studies have highlighted parental underestimation of the risks of drowning, an overestimation of their children’s ability to cope with these risks, and unsafe activity monitoring attitudes towards their children [57,60,61]. Morongliello et al. [22] highlighted that the more parents perceived their children’s progress as successful, the lower their perception of the risk of drowning, and the weaker their supervision attitudes when their offspring were near the water. These results corroborated previous work [57] showing that when parents were regularly updated on their children’s progress, they developed less of a monitoring attitude. However, parents with near-drowning experiences remained more vigilant, regardless (with or without progress feedback). Strategies to tackle this phenomenon are diverse, ranging from involving the parents in the teaching process, to the implementation of communication campaigns, information seminars, and provision of training for parents in first aid, independently from the teaching process. Results suggested that expanding “can swim” programs to enhance parents’ perceptions and understanding of children’s drowning risks and supervision needs, significantly mitigates protection against drowning. Delivering messaging in the form of ‘close-call’ drowning stories are especially effective to impact sustainable parental supervisory practices [57].

Different interventional strategies have been conducted in the literature to test their efficacy by measuring their impact on beliefs, knowledge, drowning risk, monitoring behaviours, and first aid. The first type of intervention is the use of traditional communication tools (poster, signage, information cards and fact sheets) to expose parents to information and gain their attention and interest on risks of drowning for their children [58,60]. Quan et al. [58] showed positive effects in terms of raising parents’ awareness. Other programs based on this intervention type have been inconclusive. For example, the “Keep Watch @ Public Pools” program showed that parents of children aged 0–5 years had relatively low supervision scores for attention (2.6/4), proximity to children (2.2/4), and readiness (2.8/4). This programme had no effect on these behaviours [60]. There seems to be converging evidence on the importance of including parents in learning to swim schemes in order to make them aware of drowning, as motor progress shown by children may have adverse effects. The second type of intervention consists of a more parent-focused program about water safety knowledge and attitudes [61,62] and a self-instructional first-aid program about knowledge and confidence to give Cardio-Pulmonary Resurrection [59]. Studies of direct interventions on parents (seminars, courses during children’s lessons, or first aid training) have shown positive results in terms of improving attitudes about child supervision [59,60,61,62].

Few studies have specifically examined interventions with parents (*n* = 7), although the results appear to be more homogeneous than similar studies with children. This is chiefly due to a greater homogeneity in the measurement tools used, and the variables (knowledge, beliefs, supervision) and theoretical frameworks mobilised (communication sciences and behavioural change sciences). Although the levels of evidence are still relatively low, the results are consistent in highlighting the importance of including parents in swimming classes and making them aware of the need for increased supervision, especially when children are making significant progress in aquatic skills. 

### 4.4. The Need for a Common Theoretical Framework and Tools to Promote and Assess the Aquatic Literacy of Young Children

The diversity of theoretical frameworks and measurement tools used in previous studies makes it difficult to summarise the results, particularly regarding the most relevant educational variables for teaching young children to swim. The question of “how to teach” young children, therefore, appears to be limited, not only by their quantity, but also from an epistemological point of view. While the concept of ‘water competence’ [28] has become one of the frameworks to guide programs in the question of ‘what to teach’, our results suggest that studies simultaneously addressing the effects of teaching interventions on motor, cognitive, psychological, and social aspects remain scarce. 

However, protecting children by equipping them with relevant motor skills, knowledge, and attitudes, necessary to face the dangers of drowning, while instilling the pleasure of moving in an aquatic environment, appears to be the major pedagogical challenge with regard to mitigating the tendencies towards a sedentary lifestyle in many young people [87].

The concept of aquatic literacy seems to be necessary to unify knowledge on the question of “how to teach” children of all ages. This concept, emerging from the broader concept of physical literacy, could provide the same impetus for researchers to create measurement tools and identify benchmarks to identify strengths and weaknesses in current programs. Recent studies in the field of physical literacy have shown very promising developments in the implementation of programs to educate young people to adopt an active, safe, and sustainable lifestyle [88]. The recent publication of the winners of the Erasmus Sport + program suggests that tools and benchmarks will soon be available to guide aquatic educational programs in the same way. The “ALFAC” (Aquatic Literacy for All Children) program aims to create an international database through the cooperation of researchers in seven countries (France, Lithuania, Germany, Portugal, Poland, Norway, and Belgium), based on the creation of a test battery to measure aquatic literacy. Similar to the PISA program, these benchmarks and new tools could be used as a guide to develop aquatic education programs which will allow researchers to develop common tools to facilitate the comparison and synthesis of their results.

## 5. Conclusions

The purpose of this systematic review of the literature was to examine the current state of research in the field of aquatic education in order to identify evidence-based pedagogical features to improve the early education of young children and their parents.

Our results highlighted the need to formally include parents in this process. The findings also exemplified programs which successfully developed physical, psychological/emotional, social, and cognitive dimensions of behaviour, empowering teachers with the capacity to diversify among different outcomes according to learners’ needs. However, due to the lack of intervention studies, the medium- to high-risk of bias, and the fact that none of the selected articles addressed all four dimensions of aquatic literacy simultaneously, the question of “how to teach aquatic literacy” remains open to further investigation. 

Science has two aspects. One is knowledge that has been sufficiently validated and confronted with empirical experience to be considered reliable. The questions of “what to teach” and “when to teach” seem to fall into this category. The other aspect of science is that of research, that of questions that have not yet found a robust response. The question of “how to teach” is part of this issue. We hope that there will be a mobilisation of the theoretical framework of aquatic literacy, as well as development of the tools that will be produced to educate future generations, both on the dangers of water, but also on the sheer enjoyment and pleasure to be experienced in this specific activity context.

## Figures and Tables

**Figure 1 ijerph-19-06180-f001:**
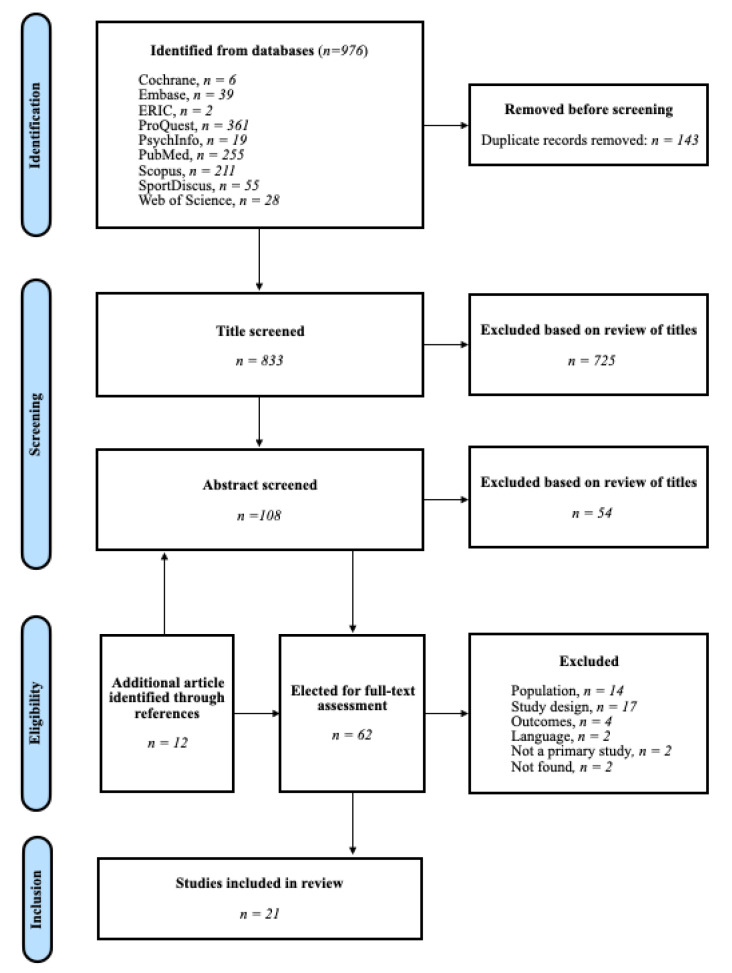
Flowchart of study selection process.

**Figure 2 ijerph-19-06180-f002:**
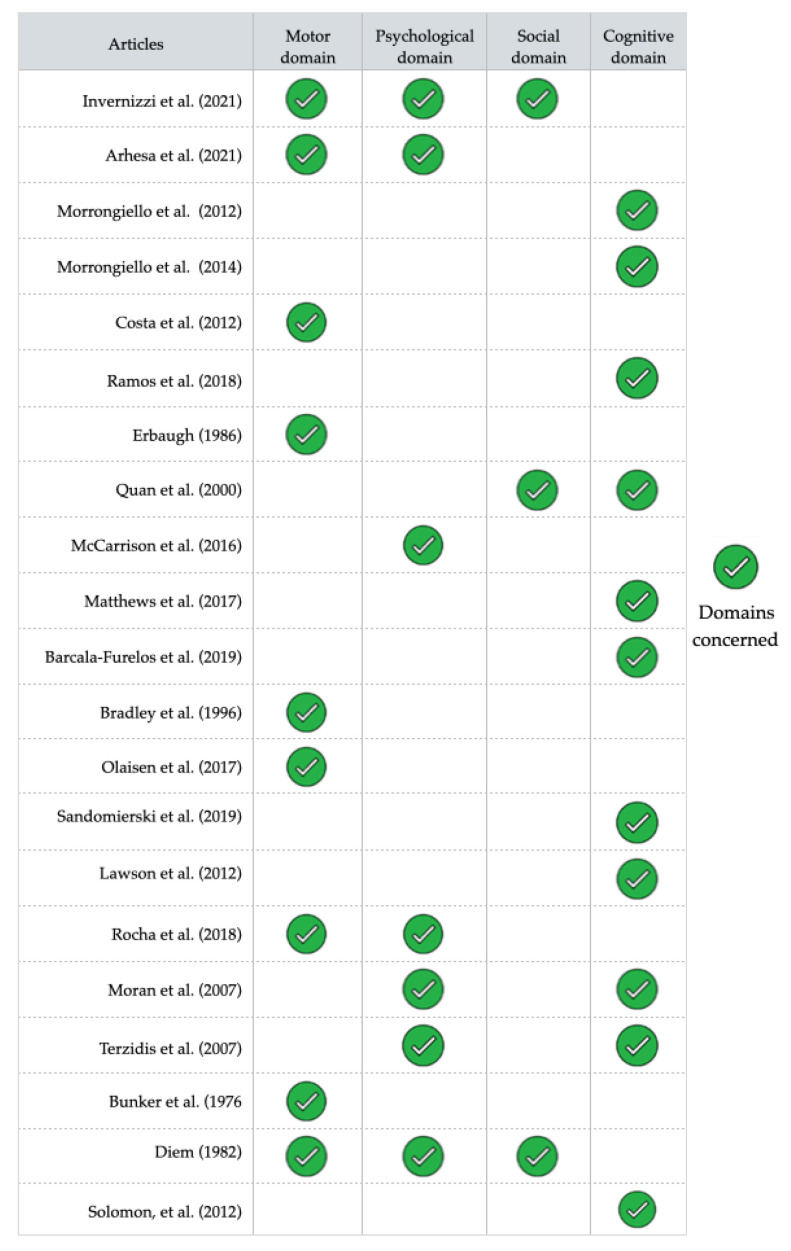
Summary of the main outcomes by Aquatic Literacy domains [22,43,44,45,46,47,48,49,50,51,52,53,54,55,56,57,58,59,60,61,62].

**Figure 3 ijerph-19-06180-f003:**
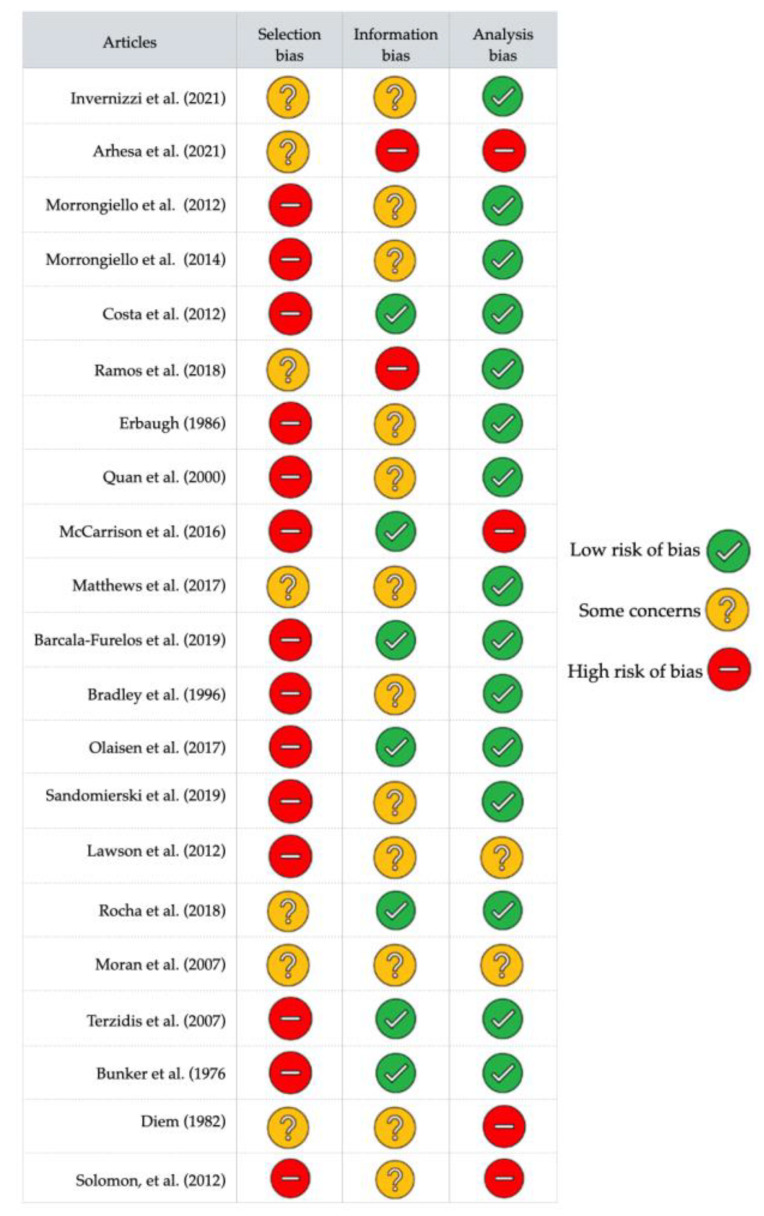
Risk of bias assessment [22,43,44,45,46,47,48,49,50,51,52,53,54,55,56,57,58,59,60,61,62].

**Table 1 ijerph-19-06180-t001:** Study using intervention towards children and their parents.

Authors, Year, Strength of Evidence [41]	Objective	Theoretical Framework(s)	Population	Study Design	Measurement(s)	Main Outcomes by AL Domains
Invernizzi et al. (2021) [43] III	Measure the effects of non-linear (NLP) vs. linear pedagogy (LP) in aquatic skills learning, and perception of children and their parents towards improvements during the training	Physical literacy [63] Grounded Theory’s model [64]	**Number** Parents: *n* = 100 (F = 53, M = 47) **Age of children** 5.9 ± 0.3 years **Groups** LP, *n* = 50 (F = 23, M = 27) NLP, *n* = 50 (F = 30, M = 20)	**Duration** 15 weeks, bi-weekly session, 50 min lessons **Pedagogy** LP vs. NLP **Children instructor ratio** 1 instructor for 8 children **Material** Shallow water	**Tools** *Parents* (1) Preliminary Interview on Swimming Course Perception [64] (2) Parent’s questionnaire about the swimming courses perception *Unvalidated questionnaire* *Children* (1) Aquatic motor competence’s test [65] (2) Pictorial scale of perceived motor competence [66] **Timing** T0: Pre-test: Preliminary interview of parents Aquatic motor competences T1: Post-test at the end of the course: Parent’s questionnaire, aquatic motor competence test Pictorial scale of perceived motor competence	**Motor** *Children significant improvement* in LP for all aquatic skills (excepted for water entry) in NLP for buoyancy, arm propulsion and recovery actions **Psychological** *Parents’ perceptions* Higher perception of children’s technical improvement and children’s confidence in LP Higher perception of children’s enjoyment in NLP Higher satisfaction towards swimming lesson for NLP *Children’ perceptions:* Higher enjoyment, self-competence in staying in deep water, and social relations for NLP

Legend: F—Female; M–Male; NLP–Non-linear Pedagogy; LP–Linear Pedagogy.

**Table 2 ijerph-19-06180-t002:** Studies using intervention towards children.

Authors, Year, Strength of Evidence [41]	Objective	Theoretical Framework(s)	Population	Study Design	Measurement(s)	Main Outcomes by AL Domains
Arhesa et al. (2021) [44] VI	Determine the effect of a playful method in swimming skills	Developmental psychology [67]	**Number** *n* = 13 (F = 8, M = 5) **Age** 4–6 years **Groups** No CG	**Duration** 36 meetings, 40 min lessons **Pedagogy** Playing method with racing games **Material** Floating and weighted object **Children instructor ratio** 6 or 7 children per trainer	**Tool** Preschool children’s swimming skill test (adapted from Susanto [67] with a qualitative scale **Timing** T1 = post-test	**Motor** Play method can improve swimming skills of preschoolers Number of children per categories: Good category, *n* = 6 Moderate category, *n* = 4 Low category, *n* = 3 **Psychological** Improvement of confidence to enter the water
Costa et al. (2012) [45] III	Analyse the differences between teaching methods in deep water (DW) & Shallow water (SW) for 4–5 years old for improving aquatic skills after 6, 12 and 18 months of practice	Aquatic motor development (Aquatic readiness, [68])	**Number** *n* = 98 **Age** 4.39 ± 0.49 years **Groups** DW, *n =* 50 SW, *n* = 48 Subgroups of 6, 12 or 18 months of swimming experience	**Duration** 6, 12 or 18 lessons, biweekly session, 40 min lesson **Environment** Deep vs. Shallow water **Children instructor ratio** 8 children per class	**Tool** *Children*: Observation table [68] of aquatic motor skills *Teachers*: Questionnaire to assess the teaching methodology [69] **Timing** Not specified but appears to be a one-time assessment at the end of the program	**Motor** No significant difference in aquatic readiness between DW and SW programs *SW group:* Seems to impose greater water competence particularly after 6 months of practice on most aquatic motor skills assessed *DW group:* After 12 months of practice the level of achievement is still significant lower in several aquatic skills *Both groups:* After 18 months of swimming practice, students have a higher number of aquatic skills acquired
Ramos et al. (2018) [46] IV	Assess the effectiveness of a classroom-based water safety program in Vietnam	Public Health and Drowning Prevention without specific theoretical framework	Quang Binh, Central Vietnam **Number** *n* = 21,043 Pre-education session, *n* = 21,043 Post-education session, *n* = 19,155 **Age** From grade 1 through 5 (approximately 5–11 years) **Groups** Grade 1 and 2 subgroup (pre educational session, *n* = 5322, post educational session, *n* = 5129)	**Duration** 90 min **Intervention type** One single on-site at schools **Program** Inspired from AUSTSWIM and Royal Lifesaving **Pedagogy** Interactive games	**Tool** Survey designed by program administrators working for an INGO focused on drowning prevention **Timing** T0 = pre-educational session T1 = post educational session (no later than within 1 week of completing the education session)	**Cognitive** Youth participants significantly increased their knowledge related to self-rescue and bystander rescue
Erbaugh (1986) [47] III	Investigate the effects of aquatic training on the swimming performance	Motor development [17]	**Number** *n* = 126 (F = 63, M = 63) **Age** 2.5–5.5 years G1, Returning program = 4.3 ± 0.7 G2, New participants = 3.6 ± 1.2 G3, CG = 3.7 ± 1.1 ***Groups*** G1, *n* = 32, with an average of 2.5 semesters of previous aquatic training G2, *n* = 30, who had no previous aquatic training G3, *n* = 64, had no previous formal swimming instruction	**Duration** 20 lessons, biweekly sessions each semester, 30 min lesson **Pedagogy** Individualised instruction using perceptual motor tasks **Material** Nontraditional equipment (hula-hoop) **Program** *Purdue Developmental Movement* Education Program **Children instructor ratio** 1 instructor for 1 child	**Tool** Erbaugh rating scale [70] assessing 6 categories of swimming tasks **Timing** T1 = 1st month T2 = 4th month T3 = 8th month	**Motor** Significant higher performances for G1 in comparison with G2 and G3 at each point in time for children’s performance of each category of tasks Aquatic training had a significant effect on the swimming performance of the G1 and group G2
Barcala-Furelos et al. (2019) [48] III	Assess a pilot childhood education program focused on the understanding, learning and memorisation of measures preventing drownings	Drowning Prevention without specific theoretical framework	**Number** *n* = 26 **Age** 5 years **Groups** CG, *n* = 12 EG, *n* = 14	**Duration** 1 week **Pedagogy** Illustrated story entitled *Xoana goes to the swimming pool and Xoana goes to the beach*	**Tool** One form for each scenario (beach and swimming pool) to assess safety and potentially hazardous elements *Not validated form* **Timing** T0 = pre-training T1 = post-training T2 = 2 months post-training	**Cognitive** Significant improvement for EG in swimming pool and beach knowledge (safety element, potential risks)
Bradley et al. (1996) [49] III	Measure performance change by 6 years old beginner swimmers participating in massed vs. distributed learning	Erbaugh [47] and Langendorfer [71]	**Number** *n* = 33 **Age** 6 years **Groups** Daily lesson, *n* = 17 (F = 8, M = 9) Weekly lesson, *n* = 16 (F = 6, M = 10)	**Duration** 10 lessons, 30 min lesson **Material** 25 m pool with cameras **Intervention type** Massed (daily lessons over 2-week period) vs. distributed (weekly lessons for 10 weeks)	**Tool** Modified Erbaugh Rating Scale-Front Crawl (MERS-F) [25] **Timing** 10 measures (one per lesson)	**Motor** Progress was similar for both groups despite the higher initial performance rating in the daily group Interval of 1 week between lessons is not detrimental to acquisition of swim strokes Front-crawl swimming skill increased significantly for both groups after the 3rd of 10 lessons No gender effect was detected
Olaisen et al. (2017) [50] IV	To evaluate the effectiveness of a swim skill acquisition intervention	Health Belief Model [72] and Social ecological framework [73]	Latinos in Redwood City, USA **Number** *n* = 149 (F = 83, M = 66) ***Age*** 3–14 years **Groups** Subgroups of 3–5 years, *n =* 44 (F = 26, M = 18)	***Parents*** **Duration** 45 min **Intervention type** One single seminar ***Children*** **Duration** 8 weeks, 1 or 2 or 3 lesson per week on the parents’ decision with a maximum of 20 lessons **Program** Learn-to-swim	**Tool** Swimming skill test *Unvalidated test* **Timing** T0 = baseline by parents’ questionnaire T1 = 4th lesson T2 = Last day of the participation	**Motor** Lesson number (more than age or gender) is the major contributing factor to the acquisition of aquatic skills A minimum of 10 lessons over 8 weeks is recommended to improve swimming skill acquisition Skills acquisition improvement was slightly high among girls
Lawson et al. (2012) [51] IV	Evaluate the impact of a water safety curriculum on safety knowledge	Public Health and Drowning Prevention without specific theoretical framework	Urban youth summer camp **Number** *n* = 166 (F = 83, M = 83) **Age** 6.9 ± 1.51 **Groups** Subgroups: Pre-K/kindergarten, *n* = 33 (F = 19, M = 14) 1st and 2nd grade, *n* = 72 (F = 31, M = 41)	**Duration** 6-week program, 4 h per week, 3 lessons for Pre-K/K group and 5 lessons for grade 1 and 2 group. **Program** *Danger Rangers Water Safety Program* (Education Adventures in collaboration with the American Association of Health Educators and Safe Kids Worldwide) **Intervention type** Water/sun safety cartoon-style video, activity book and receiving a curriculum in classroom	**Tool** Water safety knowledge test *Unvalidated test* **Timing** T0 = pre-intervention (Day 1) T1 = post-intervention (at the end of the program) T2 = 3-weeks later	**Cognitive** More water safety knowledge, better ability to list safety rules after receiving the program Significantly more rules list at T1 than at T0 for 1st and 2nd grade Pre-kindergarten/kindergarten group did not score significantly higher on the retention test compared with the pretest
Rocha et al. (2018) [52] III	Determine the effect of deep vs. shallow water differences on developing preschoolers’ aquatic skills after 6 months of practice	Aquatic Motor Skills [26]	**Number** *n* = 21 **Age** 4.7 ± 0.51 years **Groups** SW, *n* = 10 DW, *n* = 11	**Duration** 6 months, biweekly sessions, 45 min lesson **Pedagogy** Absolute control vs. guided discovery **Material** SW of 0.7 m, DW of 1.30 m Didactic-puzzles, towers, slides, mattresses, overflow arches, rings, floating-arches, balls, small boards and noodles	**Tool** Observation checklist of 17 aquatic motor skills [26] **Timing** T0 = 1st session T1 = After 6 months of practice	**Motor** *Both groups:* Improved several basic aquatic skills *SW group:* Higher degree of aquatic competence after a period of 6 months of practice *DW group:* Developed a less streamlined position at ventral gliding **Psychological** *SW group:* Association between enjoyment for swimming practice and trust about their own security in the new environment
Terzidis et al. (2007) [53] IV	Explore whether an intervention during mandatory schooling can changes water safety knowledge and attitudes	Public Health and Drowning Prevention without specific theoretical framework	Greater Athens, Greece **Number** *n* = 1400 **Age** 5–15 years Sub-group kindergarten and grade 1 pupils: 5–7 years **Groups** Kindergarten: EG, *n* = 52 CG, *n* = 115	**Duration** 1 day **Intervention type** In-class intervention **Program** Short audio-visual presentation followed by an intervention on the pupils’ comments on how relevant events could have been averted, and/or drama plays	**Tool** Knowledge and attitude with regards to water safety questionnaire *Not validated* **Timing** T0 = initial assessment T1 = post-exposure, 1 month after	**Psychological** *Kindergarten in the EG:* Improvements of attitude towards water safety and drowning prevention **Cognitive** *Kindergarten in the EG:* Improvements of attitude towards water safety and drowning prevention
Bunker et al. (1976) [54] III	Investigate the effect of video-taped FB on the learning of a continuous motor task	Cognitive development [74]	**Number** *n* = 36 (F = 18, M = 18) **Age** 4.5–6.4 and 6.5–8.5 years **Groups** Video-taped FB, *n* = 18 Auditory FB, *n* = 18	**Duration** 60 min distributed over 4 weeks **Intervention type** 15 min of correct technique for executing the flutter kick Video-taped FB vs. auditory FB **Material** Video camera	**Tool** Evaluate recorded tests based on a six-point scale *Not validated tool* **Timing** T0 = pretest T1 = posttest (5th session)	**Motor** *Auditory FB back group:* No significant improvements in flutter kicks performance *Video-taped FB group:* Performed significantly better than the groups who received traditional instruction with auditory feedback
Diem (1982) [55] IV	Assess the impact of early motor stimulation on the entire development of 4–6 years children	Psychological development [17]	Cologne, Germany **Number** *n =* 186 (F = 102, M = 87) **Age** 2.3–4 years **Groups** G1: Children who had participated in the baby swimming program from the 3rd months of life G2: Same early swimmers who received additional motor learning program from 3.6 years G3: Children who began swimming at 2.4 years Partial G4: comprised G2 Partial G5: included children who were given gymnastic training from 3.6 years Partial G6: CG	**Duration** One hour per week for 2 years **Program** Motor program to provide the child with opportunities for random movement throughout various movement planes **Material** Videotapes	**Tool** Questionnaire *Unvalidated questionnaire* 3 tests delivered 3 times *Unvalidated test* ***Timing*** T0 = beginning of the study T1, T2, T3 = during the 19-month program period	**Motor** *Motor stimulated group*: Better movement quality and accuracy, balancing and reaction **Psychological** Increase in development toward independence and self-assurance *Motor stimulated group*: Stronger development, better ability to cope with new and strange situation without the obvious effective *Undergone training and gymnastic classes group:* Better concentration *Early swimmers:* Increase their motivation performance ***Social*** *Motor stimulated group:* Greater readiness for social contact, better integration in the peer group, react more cooly to disappointments inflicted on them by their peers *Early swimmers:* Higher social behaviour
Solomon, et al. (2012) [56] IV	Determine the effectiveness of the *Whale program* which helps children from 5 to 12 to learn water safety rules	Public Health and Drowning Prevention without specific theoretical framework	Grenada ***Number*** *n =* 56 (F = 39, M = 17) ***Age*** 5–12 years Kindergarten subgroup: 5–6 years ***Group*** Subgroup of Kindergarten (*n* = 12, F = 9, M = 3)	***Duration*** 6 lessons ***Program*** *Longfellow’s WHALE Tales* program ***Intervention type*** Group discussion, posters, activities, and a video featuring an animated whale	***Tool*** Water Safety knowledge questionnaire using a pictorial scale from the WHALE Tales program *Unvalidated questionnaire* ***Timing*** T0 = pre-training T1 = post-training	** *Cognitive* ** Increase water safety knowledge *Kindergarteners* No significant effect

Legend: F—Female; M–Male; CG–Control Group; EG–Experimental Group; DW–Deep Water; SW–Shallow Water; FB–Feedback.

**Table 3 ijerph-19-06180-t003:** Studies using intervention towards parents.

Authors, Year, Strength of Evidence [41]	Objective	Theoretical Framework(s)	Population	Study Design	Measurement(s)	Main Outcomes by AL Domains
Morrongiello et al. (2013) [57] III	To examine changes in parents’ beliefs about their children’s risk of drowning, their perceived ability to swim, and their need for supervision when swimming and compare the effect of regular feedback to parents on their children’s progress and the effect of a close call of drowning	Public Health and Drowning Prevention without specific theoretical framework	**Number** T1: *n* = 387 T2: *n =* 301 **Children age** 2–5 years **Groups** FB program, *n =* 61 (T1), *n* = 45 (T2) CG, *n =* 326 (T1), *n* = 256 (T2) *In a second analysis, parents were pooled according to whether they lived a close call (39%) for drowning or not.*	**Duration** 10 weekly swimming sessions **Program** Swimming lessons towards children with: Parents receiving regular FB during about the child progress CG with parents not receiving regular FB about their child progress	**Tools** (1) Swim Ability Checklist for parents and for children (2) Drowning Prevention Beliefs Questionnaire (3) Supervision needs in outdoor drowning risk situations questionnaire *Unvalidated questionnaires* **Timing** T1 = Before the end of the 3rd lesson T2 = After the next-to-last class and before the last class	**Cognitive** *Both groups* Poor accuracy in judging children’s swimming abilities even though it improved from the beginning to the end of the swim lessons Supervision needs were underestimated and did not vary with program or change over the swim lessons Parents made more errors in judging their child’s swim ability at the T1 vs T2 *CG* Closer supervision Made more errors in judging their child’s swim ability than in the FB program *Parents who had experienced a close call of drowning* More vigilance and endorsed more watchful and proximal supervision
Morrongiello et al. (2014) [22] IV	Determine how children’s participation in swim lessons impacts parents’ appraisals of children’s drowning risk and need for supervision	Public Health and Drowning Prevention without specific theoretical framework	**Number of parents** T1: *n* = 387 T2: *n* = 301 T3: *n =* 179 T4: *n =* 119 **Children age** 2–5 years	**Duration** 8 months with around 36 lessons **Program** Non-detailed swimming program for children	**Tools** (1) Demographic questionnaire (2) Parental perception of swim ability of children (3) Parental supervision needs in near outside water scale (4) Parental perception of children ability to keep themselves safe in drowning risk situation scale *Unvalidated tools* **Timing** T1 = First 3 weeks T2 = After the next-to-last lesson and before the final lesson T3 and T4 = At the end of the last 2 lessons	**Cognitive** Perceived improvements in swim ability produce the undesirable effect of parents becoming more confident that young children can keep themselves safe near water and predicted decreased ratings of children’s supervision needs near water
Quan et al. (2020) [58] VI	Assess the effects of a drowning prevention campaign	PRECEDE-PROCEED [75,76,77] Social Marketing [76,77]	Vietnamese American Community in Seattle, USA **Number** Pre-campaign, *n* = 168 Post-campaign, *n* = 230 **Children age** 1–8 years	**Intervention type** Campaign: Key drowning prevention messages disseminated by poster, handouts, oral presentation about learn-to-swim, swim with lifeguard and wear life jacket 3 key messages: “learn to swim”,” swim with a lifeguard”, “wear a life jacket”	**Tool** Survey with 15 questions *Unvalidated survey* **Timing** T0 = pre campaign T1 = post campaign (1 year later)	**Cognitive** Significantly more respondents had heard water safety advice in the previous year Significant increase in the use of lifeguarded open water site **Social** Increased community assets: availability of low-cost family swim lessons, free lessons at beaches, low-cost life jacket sales, life jacket loan kiosks in multiple languages, and more Asian, including Vietnamese, lifeguards
McCarrison et al. (2016) [59] IV	Evaluate an evidence-based self- instructional program aimed at improving CPR knowledge and confidence	Video Self-Instruction [78]	**Number** *n* = 29 **Groups** T1, *n* = 29 T2, *n* = 29 T3, *n* = 15 Subgroup: Prior CPR education (*n =* 62.1%)	**Duration** 20 min **Program** *VSI Child CPR Program* (CPR Anytime Child of the American Heart Association) **Intervention type** One single intervention by watching the program and practicing CPR on the manikins	**Tool** Knowledge and confidence questionnaire adapted from CPR questionnaires [79,80] **Timing** T0 = Preprogram questionnaire T1 = Immediate post-program questionnaire T2 = 1-month follow-up questionnaire	**Psychological** Significant improvement regarding parental confidence in determining the need for CPR at T1 and T2 **Cognitive** Significant improvement in knowledge at T1 and T2
Matthews et al. (2017) [60] II	Examine the effectiveness of a public education program for improving child supervision levels by parents at public swimming pools	Drowning prevention and Transtheorical model of behaviour change [81]	Melbourne, Australia **Parents** *n =* 6930 IG: T0 *n* = 995/T1 *n* = 1575 CG: T0 *n =* 1925/T1 *n =* 2435 **Children** *n =* 10,186 IG: T0 *n* = 1693/T1 *n* = 2165 CG: T0 *n =* 3147/T1 *n =* 3163 **Children age** 0–14 years **Groups** Subgroups: 0–5 and 6–10 years	**Duration** 6 weeks **Program** *Keep Watch @ Public Pools* of the Royal Life Saving **Intervention type** Signage, information cards and fact sheets **Material** Videotaping	**Tool** Supervision rating scale by videotaping analysis *Unvalidated scale* **Timing** T0 = 1 week pre-intervention T1 = 1-week post-intervention	**Cognitive** *IG of 0–5 years old children* No significant effect in parental attention, proximity, and preparedness *IG of 6–10 years old children* Significant improvement in parental attention, proximity, and preparedness
Sandomierski et al. (2019) [61] III	Develop, implement, and evaluate a program targeting parents’ beliefs about children’s safety around water	Health Belief Model [82] Theory of Planned Behaviour [83] Protective Motivation Theory [84]	**Number** *n =* 242 **Age of parent’s children** 2–5 years **Groups** IG, *n =* 92 CG, *n =* 150	**Duration** 2 times 30 min lesson **Program** *S.A.F.E.R Near Water program* **Intervention type** Seminar explaining the level of supervision that is required to ensure young children’s safety around water and about supervision Additional posters **Parents instructor ratio** 1 instructor for 6 parents	**Tools** Parent Opinions About Water Safety (POAWS) Questionnaire (2) Parent Supervision Attributes Profile Questionnaire—Beach [85] ***Timing*** T0 = preintervention T1 = postintervention, 9–15 weeks later	**Cognitive** *EG* Closer supervision of their child around water Increased knowledge about children’s drowning risks and need for supervision *CG* Greater inaccuracy in their judgments related to children’s swim skill and drown-risk Greater optimism bias related to the perception that swim lessons reduce children’s need for supervision More risk in their beliefs related to supervision
Moran et al. (2007) [62] IV	Design and evaluate a pilot parent education program to improve parents’ knowledge and attitudes about water safety	Public Health and Drowning Prevention without specific theoretical framework	Auckland, New Zealand **Number** *n* = 106 **Age of children** 2–4 years	**Duration** 10 weeks **Program** Poolside safety program **Intervention type** Resources on toddler water safety while their child was receiving instruction in the pool	**Tool** Self-directed questionnaire about supervision, circumstances surrounding toddler drowning and child related CPR *Unvalidated questionnaire* **Timing** T0 = pre-intervention T1 = post-intervention	**Psychological** Improvement in parental awareness and attitudes of toddler water safety after the program **Cognitive** Increase in comprehension and awareness of the circumstances surrounding toddler drowning No significative improvement about knowledge of child CPR procedures

Legend: F—Female; M–Male; CG–Control Group; EG–Experimental Group; FB–Feedback; CPR–Cardiopulmonary Resuscitation.

## Data Availability

Not applicable.

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
