# Peer review of "Building the Foundation of Aquatic Literacy in 4–6 Years-Old Children: A Systematic Review of Good Pedagogical Practices for Children and Parents"

_ijerph, 2022, doi:10.3390/ijerph19106180_

Round 1

Reviewer 1 Report

Dear authors,

I had a pleasant moment reading your work. I hope you keep under this research path, as it is of paramount importance for the potential victims and their relatives and the future of the pedagogical practice. This sentence has no hierarchical conflict because we, as teachers, have to prevent any drowning episode through education and training. Nevertheless, any accident is regrettable.

I have found your SR very interesting. Normally I use a Checklist to facilitate my revision process and the authors’ understanding. It is attached to this revision, and some comments are in the table. Additionally, I scan the manuscript with my handwriting. You can check it, but the comments are in the checklist and the manuscript pdf. Please see the highlighting and the comments also.

I could not find any flaw that would impede your manuscript from being accepted; nevertheless, see my comments and decide with the editors and among the authors if they are suitable to make any improvement.

Abstract

The abstract contains almost all necessary components. Please check the attached file with the checklist. I will leave it to the authors to discuss with the editors if some additional information can be added. Nevertheless, the abstract adequately summarizes the main points of the article. All information presented matches that in the body of the manuscript.

I consider that the abstract will gain readers’ attention, mostly those (like me) who care about this topic.

Introduction

Congratulations on that. It was one of the best introductions I have ever read, both on revising or in a published paper. The authors provide adequate context for their topic, cite other relevant research, and go way further in framing the drowning problem (considering Stallman and the buddies' ‘paradigm shift’) deeply, not on ‘when’ or ‘what on ‘how’. In this way, the introduction clearly states the purpose of the manuscript, and I think that the general readership of the journal will find the topic meaningful.

Even though, in line 120, I had this thought in me: Aquatic Literacy… Why? Then I found myself wondering if it would be for citation purposes. Why would not water competence or aquatic competence be enough at this point since it connects with the presented frameworks and theoretical approaches? Well, the authors unveiled through the manuscript. Pardon me, please; this was just a small confession.

Methods

In the 2.1 Literature identification, when presenting the descriptors and keywords, since AL is the term you are presenting now, was it suitable for you to add aquatic/water competence?

Was there any database search in which the search process should be better described? Sometimes databases have their own ‘idiosyncrasy’!

Is the Excel Spreadsheet available for consulting? Will it be?

Results

Results are crystal clear and follow the proposed guidelines.

Comment

The discussion is relevant, and the authors discuss their findings in the context of existing research. Where appropriate, the authors discuss the relevance and importance of their findings to the specific area, identifying limitations and weaknesses.

Conclusion

The conclusion is succinct and justified by data

Figures and Tables

The information in the tables and figures is easy to interpret, being detailed enough to stand on their own, without reference to the text. Moreover, the information in the tables and figures matches the information in the text

References

Ok

Author Response

Dear reviewer, 

We, the authors, are very grateful for the time you took to review our manuscript. We were very touched by your comments and encouragement, and have tried to respond to them as well as possible.

Point 1: I had a pleasant moment reading your work. I hope you keep under this research path, as it is of paramount importance for the potential victims and their relatives and the future of the pedagogical practice. This sentence has no hierarchical conflict because we, as teachers, have to prevent any drowning episode through education and training. Nevertheless, any accident is regrettable.

Response 1: We, the authors, would like to thank you greatly for this praise. We will try to do our best to provide the clearest evidence and direction for improving aquatic education programs to reduce drowning rates and to allow as many children as possible to take opportunities to practice in the aquatic environment.

Point 2: I have found your SR very interesting. Normally I use a Checklist to facilitate my revision process and the authors’ understanding. It is attached to this revision, and some comments are in the table. Additionally, I scan the manuscript with my handwriting. You can check it, but the comments are in the checklist and the manuscript pdf. Please see the highlighting and the comments also.

I could not find any flaw that would impede your manuscript from being accepted; nevertheless, see my comments and decide with the editors and among the authors if they are suitable to make any improvement.

Response 2: We thank you for the clarity of your comments and the documents provided in the appendix. We have considered all the comments made on all the documents.

Point 3: The abstract contains almost all necessary components. Please check the attached file with the checklist. I will leave it to the authors to discuss with the editors if some additional information can be added. Nevertheless, the abstract adequately summarizes the main points of the article. All information presented matches that in the body of the manuscript.

I consider that the abstract will gain readers’ attention, mostly those (like me) who care about this topic.

Response 3: We thank you for these positive comments.

Point 4: Congratulations on that. It was one of the best introductions I have ever read, both on revising or in a published paper. The authors provide adequate context for their topic, cite other relevant research, and go way further in framing the drowning problem (considering Stallman and the buddies' ‘paradigm shift’) deeply, not on ‘when’ or ‘what on ‘how’. In this way, the introduction clearly states the purpose of the manuscript, and I think that the general readership of the journal will find the topic meaningful.

Response 4: We thank you for these supportive feedbacks.

Point 5: Even though, in line 120, I had this thought in me: Aquatic Literacy… Why? Then I found myself wondering if it would be for citation purposes. Why would not water competence or aquatic competence be enough at this point since it connects with the presented frameworks and theoretical approaches? Well, the authors unveiled through the manuscript. Pardon me, please; this was just a small confession.

Response 5: We thank for sharing his thoughts. Indeed, the concept of "aquatic literacy" is new. It comes from a scientific report commissioned by the French Ministry of Sports (refence 35), in which the authors (whose reference we have added) highlight the need to use this concept. Stallman's framework (water competence) supports the idea of thinking about swimming instruction in terms of motor skills, knowledge, and attitude. However, it does not emphasise developing the enjoyment and desire to engage in water activities in a sustainable way. For this reason, we have proposed this new concept in our introduction. 

Point 6: In the 2.1 Literature identification, when presenting the descriptors and keywords, since AL is the term you are presenting now, was it suitable for you to add aquatic/water competence?

Response 6: Thank you for this relevant comment. We have chosen to use the terms "aquatic competence" and "aquatic literacy" since the concept of "aquatic literacy" has only recently been developed in the scientific literature. Indeed, we had previously used only the term "aquatic literacy", however, the results obtained on the databases were very weak.

Point 7: Was there any database search in which the search process should be better described? Sometimes databases have their own ‘idiosyncrasy’!

Response 7: Of course, the search was not clear-cut in every database. Some of them have their own use of Boolean operators and diversification symbols.

We will provide with the new version of the article, an Excel file that will detail the search strategy for each database used.

Point 8: Method: Is the Excel Spreadsheet available for consulting? Will it be?

Response 8: Yes, as mentioned in the response of the point 7, we will provide and Excel file with the key terms used and the search algorithm for each database. 

Point 9: Results are crystal clear and follow the proposed guidelines.

Response 9: Thank you very much for this positive point.

Point 10: The discussion is relevant, and the authors discuss their findings in the context of existing research. Where appropriate, the authors discuss the relevance and importance of their findings to the specific area, identifying limitations and weaknesses.

Response 10: We thank you very much for this comment.

Point 11: The conclusion is succinct and justified by data

Response 11: We thank you for positive comment.

Point 12: The information in the tables and figures is easy to interpret, being detailed enough to stand on their own, without reference to the text. Moreover, the information in the tables and figures matches the information in the text.

Response 12: Thank you for positive feedback.

Point 13: Line 213. Check please. I guess it is figure 1!!!

Response 13: Thank you for this observation, we have rectified it at line 234.

Point 14: Table 2. Bunker et al. (1976). About the age: No sub groups? Did you handle the subjects above 6yr old the same way, or was it impossible to get the subgroup as in other studies. If so, I think you should explain in 2.2 selection process.

Response 14:

Thank you for this very pertinent comment. Indeed, for this article, no subgroup was mentioned as in the other articles. We will follow your recommendations and clarify this information in sections 2.2 and 2.3 (see lines 184, 196 and 197).

Point 15: Table 3. Quan et al. (2020) and Matthews et al. (2017). About the age: Could you differ from the parents of older than 6 children?

Response 15: Regarding the populations of these 2 articles, it was impossible to make subgroups 4-6 years. Therefore, for the Matthews article, we described the results for the 2 subgroups present in the article (0-5 years and 6-10 years).

Point 16: Line 278. I could not find this message at figure 2.

Response 16: We have reworded the sentence at line 351.

Point 17: Line 291. and 4 have a high risk

Response 17: Thank you for this observation, we have rectified it at line 636.

Point 18: Line 315-316. Carreiro da Costa. ps. Disclaimer: not me.

Response 18: Thank you for this comment, we have rectified it at line 661.

Point 19: Abstract Checklist PRISMA: Provide a structured summary including, as applicable: background; objectives; data sources; study eligibility criteria, participants, and interventions; study appraisal and synthesis methods; results; limitations; conclusions and implications of key findings; systematic review registration number: Those in bold are not present in the abstract. I am aware that there is a limitation in the word count in the abstract, but since PRISMA is used, IJERPH should make an exception in the SR and even more in SR with Meta-analysis. Up to the editors and authors.

Response 19: Thank you for pointing this out. Indeed, as you mentioned, we were annoyed about the number of words allowed in the abstract. We will make a new proposal to the editor with your advice (see from line 19 to line 22).

Point 20: Protocol and registration: No review protocol is presented. If it was registered, please provide the number and database.

Response 20: We have not registered a procedure so we cannot provide a number.

Point 21: “Information sources: Partially. Authors should inform about the date of the last database search.” Should we infer that should be the manuscript date?

Response 21: No. We have specified the date on which we ran the algorithms: between October 28th and November 15th 2021 (see line 174 and 175).

Point 22: “Search: Present full electronic search strategy for at least one database, including any limits used, such that it could be repeated” No. I am pointing out this consideration in the reviewing text.

Response 22: As mentioned in response 7 and 8, we will deliver it.

We hope that this new version will suit you. 

Sincerely yours, 

The authors

Reviewer 2 Report

This systematic review provided an interesting summary of evidence on the proposed issue, which can be very informative for practitioners and stakeholders. 

The methodology for SLR has been properly conducted, using a rigorous process.

The quality of academic and scientific writing style is very high, so the reading od the manuscript is very clear and fluent. 

I propose just minor revisions. 

Line 147. Reference 39. Authors may refer to the most recent version of PRISMA guidelines, and, in case, make their SLR according to this last version.

Check for mistakes, also highlighted in yellow: line 199, line 213, line 245, line 262, line 288, line 89. When start a sentence, write numbers with words 

Author Response

Dear reviewer, 

We, the authors, are very grateful for the time you took to review our manuscript. We were very touched by your comments and encouragement, and have tried to respond to them as well as possible.

Point 1: This systematic review provided an interesting summary of evidence on the proposed issue, which can be very informative for practitioners and stakeholders. 

The methodology for SLR has been properly conducted, using a rigorous process.

The quality of academic and scientific writing style is very high, so the reading od the manuscript is very clear and fluent.

Response 1: We, the authors, would like to thank you greatly for this praise. We will try to do our best to produce studies of this quality in this field in the future.

Point 2: Line 147. Reference 39. Authors may refer to the most recent version of PRISMA guidelines, and, in case, make their SLR according to this last version.

Response 2: Thank you for this indication that we have updated (see line 157, 1161 and 1162).

Point 3: Check for mistakes, also highlighted in yellow: line 199, line 213, line 245, line 262, line 288, line 89. When start a sentence, write numbers with words 

Response 3: Thank you for this linguistic correction. We have changed every line mentioned except lines 89 and 245 because we did not find the errors on these 2 lines (see lines 219, 234, 335 and 634).

We hope that this new version will suit you. 

Sincerely yours, 

The authors

Round 2

Reviewer 1 Report

Dear Authors,

I'm very pleased to converge with you in our thoughts. Congratulations.

Reviewer 2 Report

Authors satisfied all the requirements